# Protein Sumoylation Is Crucial for Phagocytosis in *Entamoeba histolytica* Trophozoites

**DOI:** 10.3390/ijms22115709

**Published:** 2021-05-27

**Authors:** Mitzi Díaz-Hernández, Rosario Javier-Reyna, Izaid Sotto-Ortega, Guillermina García-Rivera, Sarita Montaño, Abigail Betanzos, Dxinegueela Zanatta, Esther Orozco

**Affiliations:** 1Departamento de Infectómica y Patogénesis Molecular, Centro de Investigación y de Estudios Avanzados del IPN, Mexico City 07360, Mexico; mcdiaz@cinvestav.mx (M.D.-H.); rjavier@cinvestav.mx (R.J.-R.); gugarcia@cinvestav.mx (G.G.-R.); abetanzos@cinvestav.mx (A.B.); dxinegueela.zanatta@cinvestav.mx (D.Z.); 2Bacteriología y Laboratorio Clínico, Universidad de Santander, 200004 Valledupar, Colombia; izmcrap@hotmail.com; 3Laboratorio de Bioinformática y Simulación Molecular, Facultad de Ciencias Químico Biológicas, Universidad Autónoma de Sinaloa, Sinaloa 80030, Mexico; mmontano@uas.edu.mx; 4Consejo Nacional de Ciencia y Tecnología (Conacyt), Mexico City 03940, Mexico

**Keywords:** SUMOylation, phagocytosis, *E. histolytica*, ESCRT machinery, EhADH adhesin

## Abstract

Posttranslational modifications provide *Entamoeba histolytica* proteins the timing and signaling to intervene during different processes, such as phagocytosis. However, SUMOylation has not been studied in *E. histolytica* yet. Here, we characterized the *E. histolytica SUMO* gene, its product (EhSUMO), and the relevance of SUMOylation in phagocytosis. Our results indicated that EhSUMO has an extended N-terminus that differentiates SUMO from ubiquitin. It also presents the GG residues at the C-terminus and the ΨKXE/D binding motif, both involved in target protein contact. Additionally, the *E. histolytica* genome possesses the enzymes belonging to the SUMOylation-deSUMOylation machinery. Confocal microscopy assays disclosed a remarkable EhSUMO membrane activity with convoluted and changing structures in trophozoites during erythrophagocytosis. SUMOylated proteins appeared in pseudopodia, phagocytic channels, and around the adhered and ingested erythrocytes. Docking analysis predicted interaction of EhSUMO with EhADH (an ALIX family protein), and immunoprecipitation and immunofluorescence assays revealed that the association increased during phagocytosis; whereas the EhVps32 (a protein of the ESCRT-III complex)-EhSUMO interaction appeared stronger since basal conditions. In *EhSUMO* knocked-down trophozoites, the bizarre membranous structures disappeared, and EhSUMO interaction with EhADH and EhVps32 diminished. Our results evidenced the presence of a *SUMO* gene in *E. histolytica* and the SUMOylation relevance during phagocytosis. This is supported by bioinformatics screening of many other proteins of *E. histolytica* involved in phagocytosis, which present putative SUMOylation sites and the ΨKXE/D binding motif.

## 1. Introduction

As in other eukaryotes, in *Entamoeba histolytica*, the protozoan causative of human amoebiasis, cellular activities, including the attack to the target cell, are widely controlled by posttranslational modifications (PTMs) of proteins. Together with the perpetual movement of trophozoites, virulence expression requires intensive vesicular traffic and association to and disassociation from proteins performing the concatenated events that conduct target molecules through several compartments, for recycling or digestion. PTMs range from peptide bond cleavage and addition of phosphate and other small chemical groups, carbohydrates, and lipids to the alteration of proteins by the conjugation of modifiers such as ubiquitin and SUMO. The knowledge of the changes suffered by molecules involved in adherence to and invasion and phagocytosis of trophozoites to the target cells are pivotal to get a more comprehensive view of the parasite virulence mechanisms.

SUMO is a 10 to 13 kDa small ubiquitin-related modifier that shares 18% similarity with ubiquitin in its three-dimensional structure [1]. In the N-terminus, SUMO has an extended 10 to 25 amino acid chain, absent in ubiquitin. As this later, SUMO conjugates to its target by an isopeptide bond formed on a lysin present in the consensus sequence ΨKXE/D (ψ: hydrophobic amino acid, K: target lysin, X: any amino acid, E and D: glutamic and aspartic acids, respectively) [2]. However, many other reports indicate that alternative sequences can be used for SUMO binding [3,4,5]. Furthermore, SUMO can associate to target proteins as a single moiety or as SUMO polymers [6] in an ATP-dependent event that requires the action of E1, E2, and E3 enzymes. DeSUMOylation, the reverse process, occurs by the specific proteases UIp1a and UIp1b [7].

SUMOylation-deSUMOylation is a switch to control the cellular location of proteins and their interaction with other molecules [8]. Cell growth, differentiation, response to stress, regulation of signal transduction, gene expression, and chromatin remodeling, among others, require SUMOylation of certain proteins [9,10]. In protozoan parasites, it is known that SUMOylation takes part in cell-cycle progression and influences morphology in *Giardia lamblia* [11], while in *Trypanosoma brucei*, it contributes to chromatin organization [12] and in *Plasmodium falciparum*, participates in invasion [13].

ESCRT machinery is deeply involved in the phagocytosis of *E. histolytica* trophozoites [14,15,16]. EhVps2, EhVps20, EhVps24, and EhVps32, members of ESCRT-III complex, and EhADH, an ALIX family protein [15] and accessory member of the ESCRT machinery, participate in membrane deformation, necessary for pseudopodia and vesicle generation [17]. They are also part of the scission apparatus to form intraluminal vesicles (ILVs) in multivesicular bodies (MVBs) [17]. SUMOylation could be one of the signals for the time and place of the ESCRT proteins to act during ingestion and prsocessing of the prey, but SUMO has not been characterized in *E. histolytica*. Besides the advancement in the comprehension of the molecular events occurring during phagocytosis, the importance of understanding these phenomena relies on the possibility of carrying out the blockage of specific parasite molecules and develop better diagnostic and therapeutic methods against *E. histolytica* that infects 50 million people and kills 100,000 annually, around the world [18]. We pursued here the identification and characterization of SUMO in this parasite and investigated whether ESCRT proteins are SUMOylated during phagocytosis. Particularly, we explored the association of SUMO with EhADH and EhVps32 proteins to scrutinize the changes that they undergo, during this event. The results evidenced the active participation of SUMOylation in phagocytosis.

## 2. Results

### 2.1. In Silico Analysis Predicts the Existence of SUMO and the Sumoylation-Desumoylation Machineries in E. histolytica

To investigate whether the proteins involved in phagocytosis go through SUMOylation, we first performed bioinformatics analysis to search for *SUMO* genes in the AmoebaDB (http://amoebadb.org/amoeba/accessed on 29 April 2021), using as a template the *SUMO* sequence from *G. lamblia* [11,19]. Our search revealed two candidates in *E. histolytica*: EHI_170060 and EHI_151620. However, EHI_151620 predicts a product without the glycine residues (GG) at the C-terminus, a characteristic of SUMO [20]. In contrast, the EHI_170060 open reading frame is a 345 bp intronless sequence with 33% identity to the *G. lamblia SUMO* gene. It is located at the complementary DNA strand, between the fragments annotated as EH_170050 and EH_170070, at the 46,961 and 47,530 bp (Figure 1A). By its sequence, we estimated a protein of 12.6 kDa (EhSUMO) that has the extra amino acids at the amino terminus, recognized as the principal difference between SUMO and ubiquitin [21,22]. It has two ΨKXE/D consensus motifs at 21 to 24 and 30 to 34 residues, and displays the GG doublet at the C-terminus (112 and 113 residues), both described as SUMO interaction sites for target proteins [20] (Figure 1B). *E. histolytica,* as other protozoa [11], and *Saccharomyces cerevisiae* [23], *Drosophila melanogaster* [24], and *Caenorhabditis elegans* [25], has only one intronless *SUMO* gene, while vertebrates have four [3], and plants, eight [26].

Multiple alignments of EhSUMO amino acid sequence with SUMO of *S. cerevisiae, H. sapiens,* and *G. lamblia,* revealed 55, 48, and 33% identity, respectively (Figure 1B), and the whole gene sequence confirmed the presence of the additional bases at the amino terminus and the GG motif at the C-terminus (Figure 1B,C). 

The interactome, carried out with the putative EhSUMO as a bait and the STRING database (http://sumosp.biocuckoo.org accessed on 29 April 2021), predicted *E. histolytica* proteins that interact with SUMO to perform different functions (Table 1), already described in other systems. We identified the putative *E. histolytica* genes and proteins required for SUMOylation, to compare them with those of other organisms [7]. E1 (EHI_035540), E2a (EH1_178500), E2b (EHI_14747), E3a (EHI_0988320), and E3b (EHI_069470), presumptive SUMOylation enzymes, revealed identities from 29 to 56% to *S. cerevisiae*, 27.8 to 53.5% to *H. sapiens* SUMO-2 gene, and 25 to 49% to *G. lamblia* (Table 2).

Putative proteins in charge of deSUMOylation: UIp1a (EHI_067510), and UIp1b (EHI_097940) exhibited 23.6 to 44.3% identities to their orthologues (Table 1). The phylogenetic tree obtained using the MEGAT7 software showed that EhSUMO is close to *D. discoideum, T. cruzi,* and *Toxoplasma gondii* SUMO proteins, whereas it has a more distant phylogenetic relationship with the *H. sapiens* orthologues (Figure 2). The bioinformatics analysis strongly suggests that *E. histolytica* has a *SUMO* gene and those involved in the SUMOylation and deSUMOylation machinery.

### 2.2. The Predicted 3D-Structure of EhSUMO Is Like Other Orthologues

To confirm that the sequence that we were working with is a *bona fide* SUMO protein, we obtained its secondary and tertiary structures. The secondary structure showed the 69 amino acid chain forming the ubiquitin-2Rad60SUMO like-domain (Figure 3A), with a similar size to the one of *G. lamblia* and close to other orthologous [11,19,27,28,29]. The EhSUMO three-dimensional (3D) model, formed by a single α-helix and four β-strands, overlapped with the ones predicted for G. *lamblia* (RMSD: 0.53), and *S. cerevisiae* (RMSD: 1.26) (PDB:1L2NB), and *H. sapiens* (RMSD:1.20) SUMO-2 (PDB:1A5R) protein crystals [1,30] (Figure 3B). These findings strengthen the assumption that the EHI_170060 contig corresponds to the phylogenetically conserved SUMO in *E. histolytica.*

The EhSUMO 3D model was obtained from the I-TASSER server, selected according to its C-score and the best Ramachandran plot values (Figure 4A). After 200 ns of molecular dynamic simulations (MDS) in a soluble environment by the NAMD software, the EhSUMO 3D model conserved a single α-helix and four β-strands; the rest of the residues appeared lightly twisted in random soft coil and linear structures. The Ramachandran plot showed 98.2%, 75.7%, and 1.83% amino acids in the favored, allowed, and outside the allowed regions, respectively. Residues distribution indicated that torsion angles of certain amino acids were refined in comparison with those obtained before MDS (Figure 4A).

RMSD evaluates the system convergence during MDS and indicates whether the values follow a normal distribution [31]. EhSUMO reached the equilibrium after 100 ns (Figure 4B). The Rg values define the protein expansion and compactness. Rg revealed that EhSUMO compacted at the first 100 ns, then, it suffered an expansion from 100 to 135 ns, and in the last 60 ns of the trajectory, EhSUMO again evidenced expansion (Figure 4C), probably due to the presence of the GG region and its context. Three principal regions appeared as the most flexible areas, detected by RMSF analysis: one at M1 to N35 amino acids composed by coils and turns with non-secondary structure, explaining its higher fluctuation; the second one from Q45 to V50, in a loop, and the last one at the N-terminus from M107 to G114 residues, formed by coils and turns, with a high fluctuation (Figure 4D). Our findings confirm that the 3D model of EhSUMO has the structure predicted for other SUMO orthologues.

### 2.3. Under the Stimulus of Erythrocytes, EhSUMO Moves from the Cytoplasm to the Target Cell Adherence Points, Phagocytic Cups, and Phagosomes

According to the *in silico* data, *EhSUMO* could be a *bona fide* orthologous of *SUMO* genes, thus, we proceeded to clone the full gene and express it in *Escherichia coli*. After purification, the recombinant protein (rEhSUMO) was used to obtain rat α-EhSUMO polyclonal antibodies that in western blot assays detected a 17 kDa band in agreement to the 12.6 kDa predicted for EhSUMO plus the histidine label (Figure 5A).

Under confocal microscopy analysis, EhSUMO was located by specific antibodies and fluorescein-labeled rabbit α-rat secondary antibodies. In basal conditions, EhSUMO appeared dispersed in the cytoplasm, close to the internal plasmatic membrane, and around vesicles/vacuoles, some label was detected free in the cytoplasm, conjugated or non-conjugated to other molecules, or both (Figure 5B). After the erythrocytes stimulus was given, EhSUMO moved to the pole where the trophozoites contacted the prey; and fluorescence was more intense in the recently molded phagocytic channels. Fluorescence was also found around the ingested erythrocytes and in large phagosomes containing three or more erythrocytes (Figure 5B). Magnification of these structures revealed the bizarre figures formed in plasma and internal membranes during phagocytosis (Figure 5Ba–c). Negative controls such as pre-immune serum and only secondary antibody gave non fluorescent signals (Figure 5C). Surprisingly, western blot assays of samples obtained after different times of erythrophagocytosis did not reveal changes in the quality and quantity of proteins. The α-EhSUMO antibodies recognized at least 10 bands from around 17 to more than 240 kDa. Some of these bands might include more than one target protein or contain SUMO polymers (Figure 5D). The faint band of around 17 kDa could correspond to unconjugated EhSUMO, although Vranych and Merino [19] reported that free SUMO appeared in *G. lamblia* with higher molecular weight than the predicted one, which has been confirmed in several systems [32,33]. Intriguingly, the purified recombinant EhSUMO protein migrates at the predicted molecular weight, suggesting that inside the cell, other factors could alter its structure or migration. SUMOylation and deSUMOylation are dynamic events, and the detection of small differences through the phagocytosis kinetics might be hard. Despite this, our results evidence that under the erythrocyte stimulus, EhSUMO moves, together with certain proteins, from the cytoplasm to the phagocytic pole and vesicles, suggesting that SUMOylation could be a switch that prepares proteins to perform their role through phagocytosis.

### 2.4. In Silico Analysis Reveals Sumoylation Sites in ESCRT-III and EhADH Proteins

One of the long-term goals of our research group has been to discover events governing phagocytosis in *E. histolytica.* Therefore, we investigated if the ESCRT-III proteins and EhADH [17,34], possess sequences that make them susceptible to be SUMOylated. We employed the GPS-SUMO software [35] that predicts attraction sites for SUMO in proteins by an algorithm obtained from 983 SUMOylation sites in 545 proteins and 151 SUMO interaction motifs (SIMs) present in 80 proteins [36]. GPS-SUMO software detected putative SUMOylation sites in EhVps2, EhVps24, and EhADH sequences (T**K**LP, V**K**NE, Q**K**AA, respectively) (Table 3). Although only EhVps24 conserves the canonical SUMO-binding sequence, the three proteins have the K in the right position. In addition, according to the software, Vps20, EhVps32, and EhADH have SIMs (VTDLDQK IVDLD RQIRQNI, NNEKSHE IGDLL GEDLQDI, EYNSKAQ VILND SKKCES, respectively) (Table 3). SIMs facilitate the non-covalent conjugation of the protein to SUMO that increases its capacity for SUMOylation, altering the target protein surface and allowing its interaction with distinct molecules [37]. In addition, other proteins related to phagocytosis, such as EhVps26, EhVps35, EhCaBP1, heat shock protein 70, phosphatidylinositol phosphate kinase, EhRabB, EhRab7, EhNPCs, EhPATMK, EhABPH, myosin heavy chains 1 and Gal/GalNAc lectin heavy subunit, among others, are also predictable to be SUMOylated or interact with SUMO protein (Appendix A). These *in silico* results, predict that proteins involved in phagocytosis such as EhADH and ESCRT members are susceptible to being SUMOylated.

### 2.5. EhADH Protein Interacts with EhSUMO

In addition to the SUMOylation sites deciphered by the GPS-SUMO software [35], the secondary structure of EhADH unveiled putative SUMOylation sites at its Bro1 domain (154 amino acid) and the linker region, from 366 to 370 residues (Figure 6A). Docking modeling analysis using the 3D model of EhSUMO, obtained here, and the 3D model of EhADH previously published [31], suggested that EhADH interacts with EhSUMO by an R rich region, close to the predicted SIM site, making contact also through the S660 to V662 residues, at the C-terminus, whereas EhSUMO interacted with EhADH mainly through the N-terminus with a ΔG = −833.4 (Figure 6B).

Immunofluorescence assays using α-EhSUMO and α-EhADH antibodies, uncovered, in basal conditions, dissimilar fluorescent patterns of EhADH and EhSUMO. However, merging images revealed colocation of both proteins at pseudopodia and in regions close to the plasma membrane (Figure 6C). Immunoprecipitation assays using α-EhSUMO antibodies and trophozoites lysates confirmed this association. By western blot assays, α-EhSUMO antibodies revealed the EhSUMO protein in the input (total trophozoites proteins), and in immunoprecipitates. Similarly, the α-EhADH antibodies unveiled the EhADH protein in both samples (Figure 6D). These results strongly suggest that both proteins associate with each other, interacting directly or indirectly.

### 2.6. Colocation of EhSUMO and EhADH Increases during Phagocytosis

To investigate the cellular fate of EhSUMO and EhADH during phagocytosis, trophozoites were prepared for immunofluorescence assays and examined through confocal microscopy after different times of phagocytosis. At 5 min, EhSUMO was detected in the place of contact where erythrocytes were being ingested (Figure 7A,B), forming the peculiar membranous structures in the phagocytic channel (see Figure 5). It also appeared around ingested erythrocytes and surrounding big pockets that could be the pre-phagosomes formed to receive the red blood cells from other endosomes (Figure 7A). In some images, EhADH did not show up in the same region of the phagocytic channel that EhSUMO did, although laser sections revealed its presence in other places of this structure, as published before [38]. Large bags in the cytoplasm, that could correspond to putative pre-phagosome structures, were also enlightened by the α-EhADH antibodies, inside and surrounding them, and the label was also detected around the ingested erythrocytes (Figure 7A). At 15 and 30 min of phagocytosis, the EhADH and EhSUMO colocation increased (Figure 7), and at 60 min, both proteins moved to the internal plasma membrane and remained around the ingested erythrocytes. The erythrocytes-containing structures exhibited different sizes and numbers of erythrocytes inside. Some of them emerged stained only by α-EhSUMO or by α-EhADH antibodies, suggesting differential participation for SUMOylated proteins during the maturation of endosomes/phagosomes. The three conditions were observed: EhSUMO and EhADH separated and both proteins associated. Quantification of fluorescence colocation confirmed that it increased through the phagocytosis kinetics (Figure 7C), suggesting that the EhSUMO-EhADH association enhance during this virulence-related event.

### 2.7. In Silico Analysis Discloses Interaction between EhVps32 and EhSUMO

EhVps32 has a predicted molecular weight of 22.4 kDa, although due to its structure and charge of amino acids, it migrates at 32 kDa in SDS-PAGE [14]. Its secondary structure exhibits a SIM (NNEKSHE IGDLL GEDLQDI) from the 134 to 137 amino acids (Figure 8A), whereas its predicted 3D structure, obtained from the I-TASSER server, presents three α-helices, after 200 ns MDS by NAMD software in a soluble environment (Figure 8B). The model selected according to the C-score and the best Ramachandran plot values was employed for these analyses.

The Ramachandran plot showed 92.98%, 69.47%, and 3.16% of the amino acids in the favored, allowed, and outside the allowed regions, respectively. Besides, the RMSD analysis showed that EhVps32 reached the equilibrium at 100 ns (Figure 8C), whereas the Rg values, indicated that it compacted through the trajectory (Figure 8D), exhibiting two regions as the most flexible areas, located at E35 to K59 amino acids and the between N101-E150 residues, respectively. Docking analysis predicted that EhVps32 and EhSUMO contact each other with a ΔG = –899.24 (Figure 8F). As in the case of EhADH, the interaction was performed in a wider region than the predicted one, distributed along the whole protein. The site was composed of three N, one close to the amino terminus, the other to the C-terminus, and the third one at the RNMK motif (Figure 8F), while EhSUMO contacted EhVps32 through its N-terminus (Figure 8F). These bioinformatics data predict the association between EhSUMO and EhVps32 proteins.

### 2.8. Immunofluorescence and Immunoprecipitation Experiments Confirm the EhSUMO and EhVps32 Interaction

Confocal images of trophozoites in basal conditions, evidenced colocation of EhSUMO and EhVps32 around vesicles/vacuoles, in the inner plasma membrane, in the origin of the pseudopodia, and extensively in clumps in the cytoplasm (Figure 8G). Immunoprecipitation assays, using α-EhSUMO antibodies revealed EhSUMO and EhVps32 in the immunoprecipitates (Figure 8H). All these experiments showed an association between EhSUMO and EhVps32, although it could be direct or indirect.

### 2.9. Interaction between EhVps32 and EhSUMO Continues through Phagocytosis

Through phagocytosis, confocal images using the α-EhSUMO antibodies, disclosed the same membranous structures described above, and the small vesicles surrounding larger endosomes/phagosomes (Figure 9A,B). Whereas the α-EhVps32 antibodies appeared around the erythrocytes, in the phagocytic bags, and close to the internal plasma membrane (Figure 9A), as described [17]. Both proteins exhibited an extensive colocation, suggesting that EhSUMO is associated with EVps32 or other proteins interacting with EhVps32, which could include the ESCRT-III members and EhADH (Figure 7). However, red spots appeared around and near the red blood cells (Figure 9), showing that EhSUMO also conjugates to other proteins that are not in contact with EhVps32. In contrast to the experiments performed with α-EhADH and α-EhSUMO antibodies, colocation of EhVps32 and EhSUMO was stronger, and it was maintained and even slightly increased during phagocytosis (Figure 9A,B). Quantification of the fluorescent label in colocation confirmed this (Figure 9C).

### 2.10. EhSUMO Knocked-Down Trophozoites Exhibited Diminished Adhesion, Erythrophagocytosis and Cytopathic Effect in Comparison to the Wild Type Strain

To deepen the importance of SUMOylation in phagocytosis and other virulence properties of *E. histolytica*, we silenced the *EhSUMO* gene, using double-stranded RNA (dsRNA), expressed in bacteria, [39]. After incubation of trophozoites for 24 h with the dsRNA, lysates from silenced (KD) and control trophozoites were submitted to western blot assays. Protein patterns of control in KD trophozoites notably differed (Figure 10A). In control, bands from about 17 kDa to more than 240 kDa reproducibly appeared in the nitrocellulose membranes (like it is shown in Figure 5). However, in the KD trophozoites, bands were fainter, and some proteins of high molecular weight did not appear (Figure 10A), suggesting a lower SUMOylation of certain proteins. The antibodies against the nuclear protein EhPCNA evidenced that the amount of protein used in both lanes of the gel was similar, and degradation was not detected, at least for this protein (Figure 10A).

Polyclonal antibodies against an exposed peptide in the EhADH structure (αEhADH18; 494-KFRQFENDIKLLCEGNIQ-513 residues) (Figure 10B), recognized in the control trophozoites, the 75 kDa band of EhADH and the 112–124 kDa bands representing the distinct conformations of the EhCPADH complex. However, very faint bands were detected in KD cells (Figure 10A). The specificity of the αEhADH18 was probed by confocal microscopy assays, using pre-immune serum, or by incubating only with secondary antibodies (Figure 10C). The EhCPADH complex is formed by the association of EhADH with the EhCP112 cysteine protease [31,34] and it acquires distinct conformations, presenting slight differences in migration in SDS-PAGE. It has been reported that SUMOylation-deSUMOylation alters protein conformation, hiding relevant epitopes for protein binding and antibody recognition [40]. To explore this idea, we used two distinct polyclonal antibodies, directed against the full-length EhADH protein (α-EhADH) and another against the 18 amino acids epitope at the C-terminus of the protein. In agreement with earlier results [34,41], in the control trophozoites, α-EhADH antibodies recognized the 112–124 kDa bands corresponding to the EhCPADH complex [41], but free EhADH was not visible. However, the α-EhADH antibodies detected the 75 kDa band corresponding to the free EhADH and the EhCPADH complex (Figure 10A). In contrast, the α-EhADH 18 antibody did not react with the corresponding bands, suggesting that the epitope was not exposed. These results support the hypothesis of alteration of EhADH conformation, probably due to the lack of efficient SUMOylation. Nevertheless, this assumption needs more experiments to be precisely confirmed. Interestingly, EhVps32 appeared likewise in KD and control trophozoites, also serving as an internal control of the amount and integrity of proteins loaded in the gel. Although we would expect distinct molecular weights between the SUMOylated and non-SUMOylated EhVps32, it is possible that during the sample preparation process, the non-covalent binding of SUMO to its target could be destroyed, or that EhSUMO could be indirectly bound to EhVps32.

Next, we evaluated the effect of *EhSUMO*-silenced trophozoites in their adhesion to and phagocytosis of erythrocytes and the cytopathic effect on MDCK epithelial monolayers. Adhesion and erythrophagocytosis experiments evidenced that in all cases, KD trophozoites adhered and phagocytosed between 32 and 37% fewer erythrocytes than the wild type (Figure 11A–D). However, differences in their ability to destroy cell monolayers were lightly retarded in KD-trophozoites, in comparison with control parasites (Figure 11E,F).

Laser confocal immunofluorescence experiments verified the effect of *EhSUMO* silencing in trophozoites. The α-EhSUMO antibodies revealed a stronger signal in the control trophozoites than the KD ones. Additionally, the bizarre membranous structures formed by α-EhSUMO antibodies were not visible in KD trophozoites during phagocytosis. Besides, in contrast to the western blot assays, α-EhADH and α-EhVps32 antibodies were less reactive in the KD trophozoites by immunofluorescence assays (Figure 12A). The three proteins appeared associated in different areas in the control, even in basal conditions, but in KD trophozoites, colocation was much lower (Figure 12B,C). During erythrophagocytosis, *EhSUMO*-deficient trophozoites displayed poor colocalization and a small but significant reduction of the recognition of EhADH and EhVps32 by the antibodies (Figure 13). These findings point to the relevance of SUMOylation in phagocytosis and suggest that proteins are affected in distinct ways by the SUMOylation-deSUMOylation processes.

In conclusion, altogether, the results presented here prove the presence of an intronless *bona fide EhSUMO* gene encoding for a 17 kDa protein that actively participates in phagocytosis. Silencing of the *EhSUMO* gene affected adhesion, erytrophagocytosis, and poorly the cytopathic effect, and disturbed EhADH and EhVps32, supporting the importance of SUMOylation in phagocytosis, a landmark for the parasite virulence.

## 3. Discussion

In this paper, we disclosed and characterized the presence of a *SUMO* gene and its product in *E. histolytica.* Then, by bioinformatics analysis, we also found in the AmoebaDB the enzymes required for SUMOylation of target proteins: E1, the activating enzyme, E2, the conjugated one, and E3 the ligase, as well as those participating in deSUMOylation: UIpb1a and UIpb1b. Our data strongly suggest that SUMOylation is a modifier of the parasite proteins, stimulating some of them during phagocytosis. This assumption is supported by bioinformatics screening of many other published proteins of *E. histolytica* involved in phagocytosis, which present putative SUMOylation sites and the ΨKXE/D binding motif as shown in Appendix A**.** Furthermore, we unveiled the association of EhSUMO, a ubiquitin-like modifier (UbI) protein, with EhADH and EhVps32 proteins, both involved in phagocytosis. Beyond the detection and characterization of EhSUMO in this parasite, the relevance of this work relies on two main aspects: (i) this is the first report on the role of SUMOylation in phagocytosis and the modification of specific proteins participating in this phenomenon, and (ii) the involvement of SUMOylation during phagocytosis highlights the potential use of this knowledge for the development of therapeutics and diagnosis methods to defeat amoebiasis.

Lack of vaccines, reduced chemotherapy options, and the emergence of drug-resistant parasites [42] are challenges presented by diseases caused by protozoa, among these, amoebiasis. PTMs, including UbIs, modify proteins to facilitate their functions, including virulence-related functions (43). UbIs have been widely studied in yeast, plants, and Mammals [43,44], but little is known on their role in protozoan parasites, highly divergent organisms with characteristics that might be investigated to understand their evolutionary process and their virulence mechanisms [45,46]. Ubls impact the regulation of cellular functions such as cell cycle progression [11], transcription [47,48], stress responses [49], DNA damage repair [50], cell signaling [22], nuclear transport [51], and autophagy [43]. By findings reported here, we add to this list, an old event, found now to be regulated by SUMO: the phagocytosis, involved in damage produced by *E. histolytica* trophozoites.

In *E. histolytica*, Arya et al. in 2012 [52] reported a bioinformatics study on UbIs modifiers and their conjugated enzymes, and recently, Kumari et al. In 2018 [53] found a UBc7/Ube2g2 protein connected with the plasma membrane and phagocytosis in trophozoites. Besides this, we have not found reports on SUMOylation in this parasite. By *in silico* analysis, we detected EHI_170060 contig that presents the characteristics described so far for SUMO. In the phylogenetic tree, EHI_1700060 product (*EhSUMO*) appeared close to *T. cruzi* and *D. discoideum* orthologues. However, we used here as templates the yeast and human SUMO-2 proteins, because they have been extensively studied, their 3D structures are well known, and crystal is available. HsSUMO-2 forms stable polymeric chains that also are susceptible to poly-ubiquitination, a signal for proteasome degradation [54]. This effect is given by the K11 of the ΨKxE/D consensus motif that allows the formation of poly-SUMO chains, absent in HsSUMO-1. Although we have not studied its relevance of this K in trophozoites, EhSUMO possesses the K amino acid at the motif ΨKxE/D, but at K22. These facts and others, including the MDS analysis [1,55], support the identity and nature of EhSUMO.

Docking of EhADH and EhSUMO, and EhVps32 and EhSUMO, predicted that EhSUMO uses different motifs in the protein target to join to other proteins, but targets suggested by our results could need fine-tuning of punctual mutations to precisely determine the binding sites. However, the confocal images evidenced that EhADH and EhVps32 efficiently bind, directly or indirectly, to EhSUMO, and that SUMOylation influences their cellular location, although we cannot discern how much of the changes were due to the phagocytosis event and how much to the SUMOylation process. Nevertheless, the fact that KD trophozoites in basal conditions did not display EhADH close to the plasma membrane or in pseudopodia, suggests that EhADH needs to be SUMOylated to find its cellular position. Confocal assays using control and KD trophozoites, support this assumption. Altogether, these results point out the hypothesis that the equilibrium between PTM proteins and their unmodified state modulates the fate and function of the target substrates during phagocytosis, granting the fine-tuning of the cellular mechanisms needed for life.

Except for EhADH and EhVps32, we could not disclose changes in the amount and nature of the proteins that are SUMOylated during phagocytosis, but SUMOylation-deSUMOylation is a highly dynamic process, and undetectable changes could be occurring. The multiple and speedy changes produced during this phenomenon were evidenced by the images obtained using α-EhSUMO antibodies that uncovered the active participation of EhSUMO. The images revealed the extensive membrane changes accordingly to the moment of erythrocyte’s contact and ingestion during the intense vesicular trafficking accelerated by phagocytosis. We did not find a single repetitive bizarre figure in a trophozoite or the total population, corroborating the dynamics of the event. SUMOylation alters not only the cellular location of the target protein, but changes the protein conformation, and consequently, their affinity to other proteins [56]. This has been studied mainly in mammalian cells and yeast, but little in parasites, even when these events might be part of their virulence mechanisms.

The change of EhADH conformation was suggested by the lack of recognition by the α-EhADH18 antibodies in KD trophozoites. To discard protein degradation, we used a polyclonal antibody directed to the full-length EhADH protein that detected the complex in both types of trophozoites and reacted with the 75 kDa band, evidencing the integrity of the protein. We suspect that the SUMOylation could produce stability on EhADH, but when EhSUMO is diminished, the non-SUMOylated protein could change its structure. MDS experiments characterizing the different protein conformations showed that EhADH assumes distinct structures [31]. *EhSUMO* silencing trophozoites showed a diminished capacity to adhere and ingest red blood cells, plausible due to a poor SUMOylation of certain proteins. However, the cytopathic effect assays evidenced that the KD-trophozoites were only poorly affected, confirming that distinct proteins are participating in adhesion, phagocytosis, and cellular destruction.

The position of EhADH and EhVps32 inside the cell, after they were hypothetically SUMOylated (deduced by the colocation images), constantly change during phagocytosis. Each protein behaved differently: colocalization of EhADH and EhSUMO significantly increased through phagocytosis, whereas EhVps32 and EhSUMO maintain a high level of colocalization since basal state. These differences could be interpreted as necessary events to carry out the distinct functions of the proteins. Our work with EhADH since many years ago [41], has evidenced its interaction with many other molecules, such as EhVps32 [14] and EhNPC1 and EhNPC2 [57] as it has been described for other ALIX family proteins [58], thus, it is possible that the protein conformation could be altered according to the target protein. Moreover, these four proteins presented putative SUMOylation sites in our *in silico* screening. Meanwhile, the EhVps32 movement is probably limited to the ESCRT-III complex participation. Hence, SUMOylation could enable them to move in the cell and perform their functions.

Karpiyevich and Artavanis-Tsakonas in 2020 [44] postulated that in addition to fulfiling the conserved functions described for these modifiers, Ub1s participate in novel parasite-specific roles. The data presented here, support the Karpiyevich and Artavanis-Tsakonas hypothesis [44], suggesting that proteins involved in phagocytosis of *E. histolytica* trophozoites suffer SUMOylation, as a requisite to carry out their tasks. In conclusion, our findings point out the importance of SUMOylation in phagocytosis. Based on these data, it is possible that in the future, inhibition of SUMOylation in *E. histolytica* and in other parasites could help to find novel therapeutic methods to defeat amoebiasis and other parasitic diseases.

## 4. Materials and Methods

### 4.1. Culture of Trophozoites and Epithelial Cells

Trophozoites of *E. histolytica,* clone A (strain HM1: IMSS) [59], were axenically cultured in TYI-S-33 medium at 37 °C and harvested in a logarithmic growth phase for all experiments [60].

Madin Darby canine kidney (MDCK) epithelial cells were grown in DMEM medium (Gibco) supplemented with 100 IU/mL penicillin (in vitro), 100 mg/mL streptomycin (in vitro), 10% fetal bovine serum (Gibco), and 0.08 U/mL insulin (Eli Lilly), at 37 °C in a 95% air and 5% CO_2_ atmosphere.

### 4.2. SUMO Searching and Phylogenetic Analysis

Full-length sequence of *SUMO* gene was identified in the *E. histolytica* genome, using the BLASTP algorithm (http://blast.ncbi.nlm.nih.gov//Blast.cgi accessed on 29 April 2021) and the AmoebaDB database (www.amoebadb.org accessed on 29 April 2021). Additionally, SUMO proteins of *G. lamblia* (accession number GL50803_7760)*, H. sapiens* (accession number NP_001005781), and *S. cerevisiae* (accession number NP_010798.1) were also explored. The *EhSUMO* sequence was analyzed and 5’ and 3’ end primers were designed, to amplify the full-length gene. The putative amino acid sequence of EhSUMO (EHI_170060) was aligned with its orthologous by ClustalW and data were submitted to phylogenetic analysis by UPGMA, using MEGA 5.05 software. The bootstrapping was performed in 1000 replicates.

### 4.3. Secondary and Tertiary Structure of EhSUMO

By *in silico* analysis, the ubiquitin-2 Rad60 domain, characteristic of SUMO proteins, was located and aligned in the putative EhSUMO amino acid sequence and compared with SUMO orthologous, to design the secondary structure of the protein. The predicted 3D structure of EhSUMO was obtained using the crystal of *H. sapiens* (PDB:1A5R) and *S. cerevisiae* (PDB:1L2N). To obtained the predicted 3D structure of *G. lamblia* SUMO, we submitted the protein sequence V6TGL retrieved from UniProtKB, to the I-TASSER server and the higher C-score was selected. The structural alignment of proteins was visualized with VMD [61].

### 4.4. Molecular Dynamics Simulations (MDS)

To predict the interaction between EhSUMO and EhVps32 or EhADH proteins, we performed molecular dynamics simulations (MDS) of the predicted 3D model of EhSUMO and EhVps32 and we took the published data for the 3D model of EhADH [31]. The 3D model of EhSUMO and EhVPS32 was obtained from their amino acid sequences using ID C4M1C8 and C4M1A5, respectively (uniprot.https//www.uniprot.org accessed on 29 April 2021), and the I-TASSER server (https://zhanglab.ccmb.med.umich.edu/I-TASSER accessed on 29 April 2021). The crystallographic structures that I-TASSER used to obtain the 3D EhSUMO model were: 5GJL solution structure of *Plasmodium falciparum*, IWZ0 solution structure of human SUMO-2 (SMT3B), a ubiquitin-like protein 5XQM structure of SMO1, SUMO homolog of *Caenorhabditis elegans*, and 2K8H solution structure of SUMO from *Trypanosoma brucei.* Those used for the EhVps32 model were: 5FD7 and 5FD9 crystal structures of ESCRT-III Snf7 core domain (conformation A and B, respectively) from *Saccharomyces cerevisiae*, 5NNV structure of a *Bacillus subtilis* Sms coiled-coil middle fragment, 2GD5 structural basils for budding by ESCRT-III factor ChMP3 from *H. sapiens* and 5NL crystal structure of the two spectrin repeated domains from *E. histolytica.*

MDS were carried out using the NAMD 2.8 software [62], through GPU-CUDA with video card graphics NVIDIA Tesla C2070/Tesla C2075. The force fields CHARMM22 and CHARMM27 [63] were used to create the topologies for protein and lipids, respectively. The TIP3 model was employed for water molecules. The system was solvated by the *psfgen* software in the VMD program [61]. By this software, the NAMD program added water molecules and sodium ions to neutralize the system: one sodium ion was added to EhSUMO with 29,329 water molecules, and 14 sodium ions with 8719 water molecules to neutralize the system for EhVps32. Both systems were minimized for 1000 steps followed by equilibration, under constant temperature and pressure (NPT) conditions for one ns with protein and lipid atoms restrained. Afterward, 200-ns-long MDS was run, considering EhSUMO and EhVPS32 proteins as soluble, without position restraints under periodic boundary conditions, and using an NPT ensemble at 310 K. Particle mesh Ewald technique was calculated for the electrostatic interactions method [64]. Nine Å cut-off was used for the van der Waals interactions. The time step was set to 2.0 fs, and the coordinates were saved for analyses every one ps; 200 ns of MDS was carried out for both proteins, then, protein-protein docking calculations were performed using different conformers through 200 ns of MDS. Simulations were performed in the Laboratory of Molecular Modelling and Bioinformatics of the Facultad de Ciencias Químico Biológicas de la Universidad Autónoma de Sinaloa and the Hybrid Cluster Xiuhcoatl (http://clusterhibrido.cinvestav.mx accessed on 29 April 2021) of CINVESTAV-IPN, México.

The structural analysis from the MDS, the root mean square fluctuation (RMSF), and the radius of gyration (Rg), as well as the snapshots used for docking analysis, were obtained and analyzed with the grcarma software [65]. Root mean square deviation (RMSD) was normalized using the SigmaPlot 12.0 software. The protein–protein predicted docking studies were calculated employing different conformers with Cluspro server [66,67]. Molecular graphics were performed in SigmaPlot 12.0 and all 3D-structures visualization was performed by VMD [61].

### 4.5. Cloning of the E. histolytica SUMO Gene (EhSUMO)

To clone *EhSUMO* gene, the full DNA sequence of 345 bp was PCR-amplified, using the following specific primers: sense 5’-CCGGTACCATGTCTAATCAACCACAATATGGAATTAAATC-3’ and antisense 5’-CCGGATCCTTATTTGATGTATTGAAGGTATTGAGTATTAAAAAGA-3’, in a mixture containing 10 mM dNTPs, 100 ng of *E. histolytica* genomic DNA or cDNA as template, and 2.5 U of *Taq* DNA polymerase (Gibco). PCR assay was carried out for 35 cycles (1 min at 94 °C, 30 sec at 59 °C, and 40 sec at 72 °C.) The sense oligonucleotide contains a *KpnI* restriction site, while the antisense oligonucleotide harbors a *BamH1* restriction site. The full-length gene was cloned in pCold II DNA plasmid, which conferred a histidine tag [68]. As a positive control of the reaction, we use primers to amplify the *Ehgata* gene [69]. Two negative controls were included: in the first, water was used instead of template; and in the second, the DNAse was omitted in the mixture.

### 4.6. Expression and Purification of Recombinant Protein, and Generation of Anti-Ehsumo Antibodies

*Escherichia coli* BLI21 (pLysS) bacteria were transformed with the pCold/*EhSUMO*, containing the full open reading frame of the *EhSUMO* gene to produce a His-tagged EhSUMO recombinant protein (rEhSUMO). The His-rEhSUMO protein was purified with cobalt beads in an imidazole gradient and used to subcutaneously and intramuscularly inoculate Wistar rats (50 µg emulsified in Titer-Max Classic adjuvant, 1:1) (Sigma), to generate α-EhSUMO polyclonal antibodies. Two more doses (50 µg) were injected at 15-days intervals, and animals were bled to obtain the immune serum. Pre-immune serum was also obtained, before immunization.

### 4.7. Production of the α-EhADH18 Antibody

The KFRQFENDIKLLCEGNIQ peptide from EhADH (495 to 512 amino acids) was synthesized together with the KLH (Keyhole Limpet Hemocyanin) tag to increase its immunogenicity (GenScript, Piscataway, NJ, USA). New Zeland rabbit (already-existing collection) was immunized with 600 μg of the peptide resuspended in TiterMax® Gold adjuvant (1:1) (Sigma, St. Louis, MO, USA) and then, three more immunizations were performed at 15-day intervals, followed by bleeding to obtain the antibody named anti-EhADH18. Pre-immune serum was also obtained, before immunizations.

### 4.8. Western Blot Experiments

Trophozoites lysates (35 µg), or purified rEhSUMO were electrophoresed in 15% sodium dodecyl sulfate-polyacrylamide gels (SDS-PAGE), transferred to nitrocellulose membranes and probed with rat α-EhSUMO (1:2000), mouse α-EhVps32 (1:1000), rabbit α-EhADH (1:1000) [34], rabbit α-EhCPADH (1:35,000) [41], mouse α-actin (1:3000) kindly donated by Dr. José Manuel Hernández (Cell Biology Department, CINVESTAV) and mouse α-EhPCNA (1:500) antibodies [70]. Membranes were washed, incubated with α-rat, α-mouse, and α-rabbit HRP-labeled secondary antibody (Sigma, 1:10,000), and revealed with ECL Prime detection reagent (GE-Healthcare, Chicago, IL, USA), according to the manufacturer´s instructions.

### 4.9. Laser Confocal Microscopy Assays

Trophozoites were grown on coverslips, fixed with 4% paraformaldehyde at 37 °C for 1h, permeabilized with 0.5% Triton X-100 and blocked with 10% fetal bovine serum in PBS. Then, cells were incubated at 4 °C for overnight (ON) with either α-EhSUMO (1:100) or α-EhVps32 (1:100) antibodies labeled with Alexa-555 or Pacific Blue kit (Molecular Probes-Thermo Fisher), respectively, or with rabbit α-EhADH (1:100) antibodies. After extensive washing, samples were incubated for 30 min at 37 °C with α-rabbit FITC-labeled secondary antibody (1:100). Fluorescence was preserved using Vectashield antifade reagent (Vector), examined through a Carl Zeiss LMS 700 confocal microscope, in laser sections of 0.5 μm and processed with ZEN 2009 Light Edition Software (Zeiss). To evaluate the colocation between proteins, fluorescence intensity was quantified from at least 30 confocal images, using the ImageJ 1.45v software and the JACoP plugin.

### 4.10. In Vivo Virulence of E. histolytica Trophozoites

Trophozoites were incubated at 4° and 37 °C with erythrocytes (1:25 ratio) for different times for adhesion and phagocytosis assays, respectively [41]. For some experiments, ingested and adhered trophozoites were contrasted by Novikoff staining [71]. After phagocytosis experiments, samples were processed for immunofluorescence and western blot assays [59]. Three independent experiments were performed and the number of erythrocytes adhered or ingested per trophozoite was obtained from 100 trophozoites.

For cytopathic assays, trophozoites were incubated with confluent MDCK cell monolayers (1:20 ratio) for 1 h. After this time, the parasites were removed by several washes with cold PBS, the remaining epithelial cells were fixed with 2.5% glutaraldehyde and stained with 1% methylene blue for 10 min. The MDCK cell destruction was represented in comparison with control cells not incubated with trophozoites. Dye concentration was quantified using the ImageJ software [41].

### 4.11. Immunoprecipitation Assays

Trophozoites were lysed in the presence of 10 mM Tris-HCl, 50-mM NaCl, and proteases inhibitors, by freeze-thawing cycles and vortexing. Immunoprecipitation assays were performed using 200 μL of protein G-agarose (Invitrogen, Carlsbad, CA, USA) and α-EhSUMO antibody as described [14]. Immunoprecipitates were analyzed by western blot assays using α-EhSUMO, α-EhADH α-EhVps32, and α-actin antibodies, as described above.

### 4.12. Silencing Assay

The knock-down (KD) of *EhSUMO* was performed using the bacterial expression of double-stranded RNA (dsRNA) and parasite soaking experiments as described [39]. Briefly, the 345 bp from the 5’-end of the *EhSUMO* gene were PCR-amplified using the following primers: sense 3’-CCGAGCTCATGTCTGACGCACAACATTCA-5’ and antisense 3’-CCGGTACCTTAAAACCCACCAACTTGATTCAT-5’. Then, the amplicon was cloned into pJET1.2/blunt plasmid and subcloned into pL4440 plasmid, using the *Sac1* and *Kpn1* restriction sites. PCR, restriction analysis, and DNA sequencing were performed to verify the resulting pL4440-*EhSUMO* plasmid. The competent RNase III-deficient *E. coli* strain HT115 was transformed with the pL4440-*EhSUMO*. Bacteria were grown at 37 °C in LB broth for plasmid construction or 2YT broth for dsRNA expression, in the presence of ampicillin (100 mg/mL) and tetracycline (10 mg/mL) [72]. The expression of *EhSUMO*-dsRNA was induced with 2-mM isopropyl β-D-1-thiogalactopyranoside ON at 37 °C. Then, bacterial pellet was mixed with 1 M ammonium acetate and 10 mM EDTA, incubated with phenol:chloroform:isoamyl alcohol (25:24:1) and centrifuged. The supernatant was mixed with isopropanol, centrifuged, and the nucleic acid pellet was washed with 70% ethanol. DNase I (Invitrogen) and RNase A (Ambion, Austin, TX, USA) were added to eliminate ssRNA and dsDNA molecules. *EhSUMO*-dsRNA was washed again with isopropanol and ethanol, analyzed by agarose gel electrophoresis, and concentration determined by spectrophotometry. Lastly, purified *EhSUMO*-dsRNA (10 μg/mL) molecules were added to trophozoites (3.0 × 10^4^) in TYI-S-33 complete medium, and cultures were incubated at 37 °C for 24, 48, 72, 96, and 120 h. At 24 h was the time when silencing of the EhSUMO protein was visualized and analyzed by western blot assays and confocal microscopy. All subsequent experiments were done at this time. Trophozoites growing under standard conditions (without dsRNA), were used as controls.

### 4.13. Statistical Analysis

Statistical analyses were performed by t-Student test, using GraphPad Prism 5.0 software. The scores showing statistically significant differences are indicated with asterisks in the graphs. The corresponding *p* values are indicated in figure legends.

## 5. Conclusions

Phagocytosis is one of the main functions that *Entamoeba histolytica* trophozoites carry out during the invasion of the host. Many proteins are involved in this fascinating event, in which the plasmatic membrane undergoes multiple and speedy changes. Postraduccional modifications activate proteins in the precise time that they must get involved. SUMOylation, which consists of the non-covalent binding of SUMO protein with target molecules, is one of the main changes suffered by proteins to enable them to participate in cellular functions. SUMOylation had not been studied in *E. histolytica* nor phagocytosis, and our working hypothesis is that this event is deeply engaged in the ingestion of target molecules and cells. The results of this paper prove the presence of an intronless bona fide *EhSUMO* gene encoding for a 17 kDa protein that is actively involved in phagocytosis. Silencing of the *EhSUMO* gene affected the rate of adhesion and phagocytosis and to a lesser extent the cytopathic effect and interfered with the EhADH and EhVps32 function, two proteins involved in phagocytosis, strongly supporting the importance of SUMOylation in this event.

## Figures and Tables

**Figure 1 ijms-22-05709-f001:**
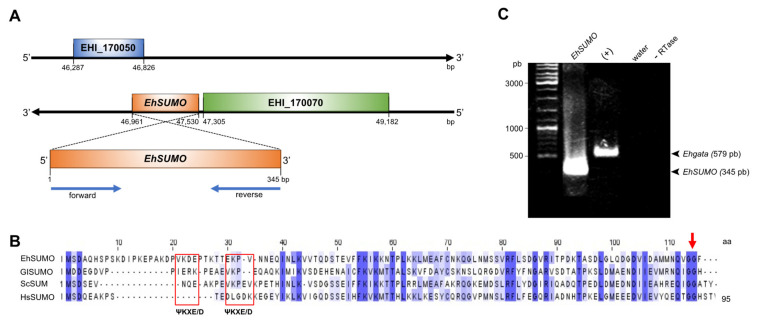
Identification and location of the *EhSUMO gene* in *E. histolytica genome*. (**A**) Location of *EhSUMO* gene in the genome. *EhSUMO* is in a contig located between 47,530 to 46,961 bp in the DNA negative chain flanked by two hypothetic genes (EHI_170050 and EHI_170070). Blue arrows: primer designed for *EhSUMO* amplification. (**B**) Comparative alignment of predicted EhSUMO amino acid sequence with proteins from other organisms, the blue color indicates the identity and similarity of amino acids. Red squares signal the ΨKXE/D motifs and arrow the GG doublet. (**C**) PCR amplification using complementary DNA (cDNA) as a template and specific primers for *EhSUMO. Ehgata*: positive control (+). Negative controls: in the mixture instead of template water was added or in the cDNA production, the RTase was omitted (-RTase). Arrowheads: PCR products.

**Figure 2 ijms-22-05709-f002:**
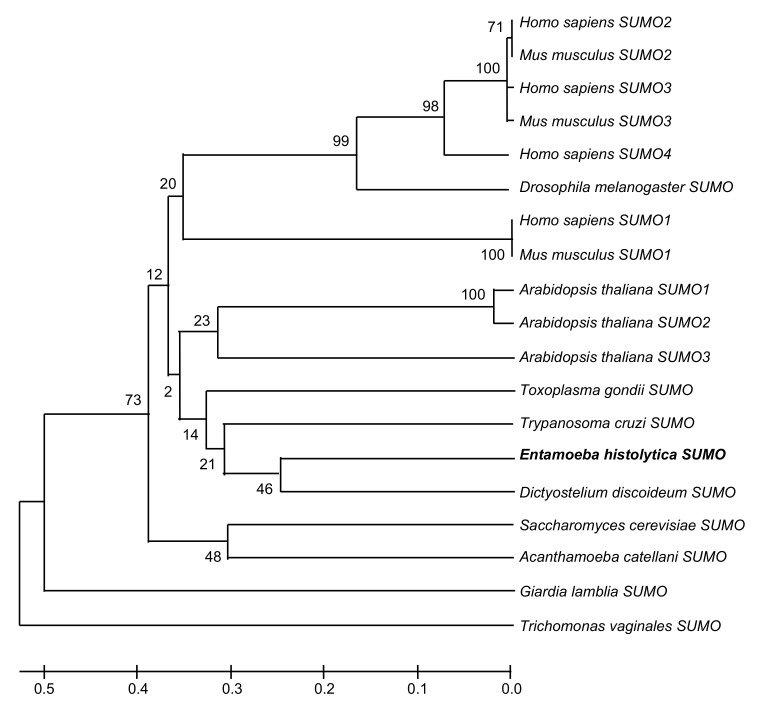
Phylogenetic analysis of EhSUMO. Phylogenetic tree performed by UPGMA, using MEGA 5.05 software, shows the position of *E. histolytica* SUMO protein among different species. Numbers in horizontal lines indicate the confidence percentages of the tree topology from bootstrap analysis of 1000 replicates.

**Figure 3 ijms-22-05709-f003:**
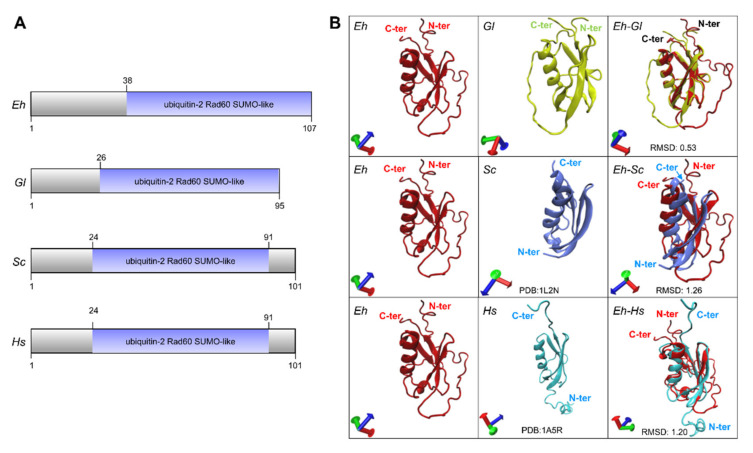
Secondary and tertiary structures of EhSUMO and its orthologous. (**A**) Schematic comparison of functional domains of EhSUMO and other SUMO proteins from different systems. Numbers at the right correspond to the amino acids forming each protein. (**B**) 3D-model of EhSUMO protein (*Eh*), overlapped with those of *G. lamblia* (*Gl*), *S. cerevisiae* (*Sc*), and *H. sapiens* (*Hs*). The N- and C-terminus regions are signaled in red letters. At the bottom of each square are indicated the predicted 3D structures used for *G. lamblia* and *H. sapiens.*

**Figure 4 ijms-22-05709-f004:**
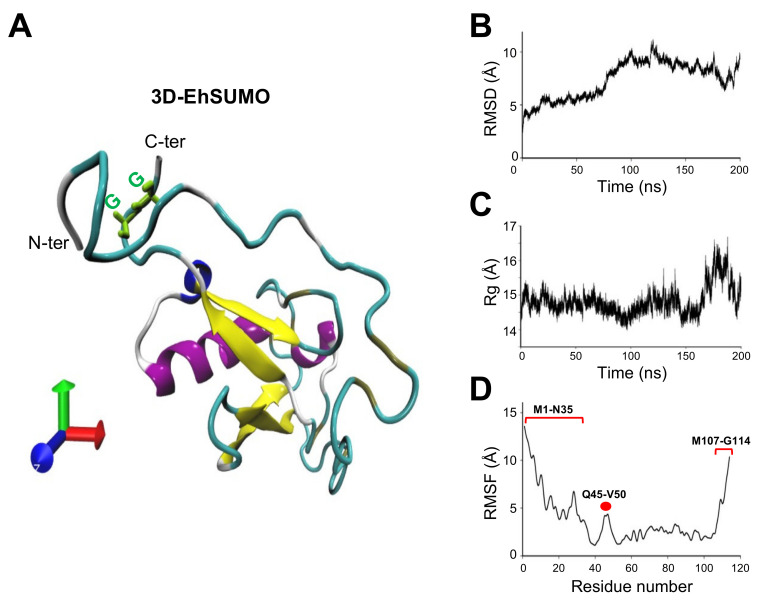
Refined the predicted 3D structure of EhSUMO and MDS. (**A**) Model of EhSUMO presenting the best C-score after 200 ns of MDS. The N- and C-terminus regions, as well as the GG residues, are indicated. (**B**–**D**) The structural analysis of MDS was carried out by RMSD (**B**), the radius of gyration (**C**), and RMSF (**D**). The most flexible regions are indicated in red dot and brackets.

**Figure 5 ijms-22-05709-f005:**
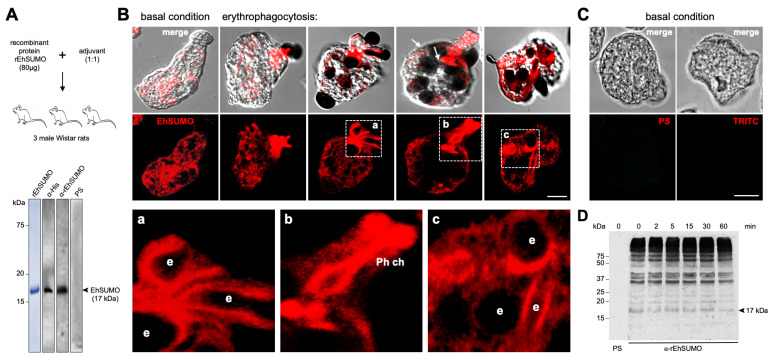
Cellular location of EhSUMO during phagocytosis. (**A**) Immunization scheme to produce α-EhSUMO antibodies, probed in western blot assays, using bacterial lysates. α-His antibody: positive control. Pre-immune serum (PS): negative control. Lane 1: rEhSUMO stained with Coomassie blue. Numbers at left: molecular weight standards. (**B**) Confocal microscopy of trophozoites in basal condition and after erythrophagocytosis, using α-EhSUMO antibodies. Squares: different structures formed during phagocytosis magnified in (**a**–**c**). (**a**) Structure with the form of a conduit or channel. (**b**) Phagocytic channel (Ph ch). (**c**) The elongated structure surrounding a phagocytosed erythrocyte. e: erythrocytes. Scale bar = 10 μm. (**C**) Negative controls: trophozoites in basal conditions incubated with PS or only with the secondary antibody (TRITC). (**D**) Western blot analysis of trophozoites lysate in basal condition (0 min) and after erythrophagocytosis, using PS or α-EhSUMO antibodies. Number at left: molecular weight standards.

**Figure 6 ijms-22-05709-f006:**
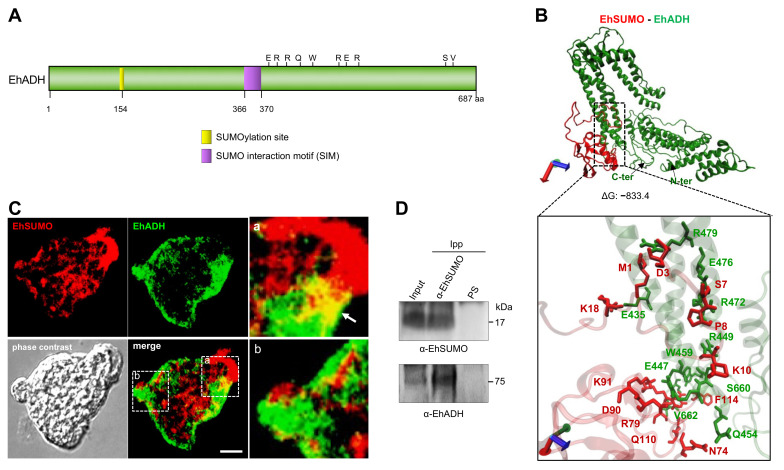
Association of EhSUMO and EhADH. (**A**) Schematic representation of EhADH protein with SUMOylation (yellow) sites and SIM sites (purple). (**B**) Molecular docking between predicted EhSUMO (red) and predicted EhADH (green). Square in the bottom: magnification of the interacting residues. Axes: x, red; y, green; z, blue. (**C**) Immunofluorescence assays of trophozoites under basal conditions using α-EhSUMO (red) and α-EhADH (green) antibodies. Regions in squares are magnified. Arrow: colocalization area. Scale bar = 10 µm. (**D**) Western blot of Immunoprecipitates using trophozoites lysates and α-EhSUMO antibody or pre-immune serum (PS) and probed with α-EhSUMO or α-EhADH antibodies. Numbers at right: molecular weight of immunodetected proteins.

**Figure 7 ijms-22-05709-f007:**
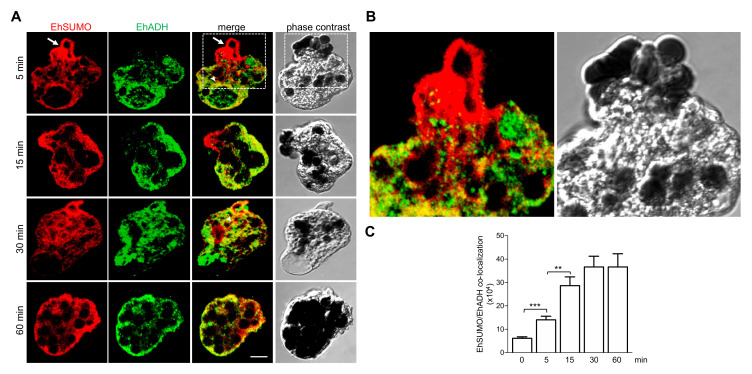
Location of EhSUMO and EhADH during phagocytosis. Trophozoites were incubated Figure 5. 15, 30, and 60 min with erythrocytes and processed for confocal microscopy. (**A**) Immunofluorescence assays using α-EhSUMO (red) and α-EhADH (green) antibodies. Arrow: EhSUMO in erythrocytes being phagocytosed. Scale bar = 10 µm. (**B**) Magnification of squares in (**A**). (**C**) Quantification of EhSUMO and EhADH colocalization in the whole cells. (**) *p* < 0.01 (***) *p* < 0.001.

**Figure 8 ijms-22-05709-f008:**
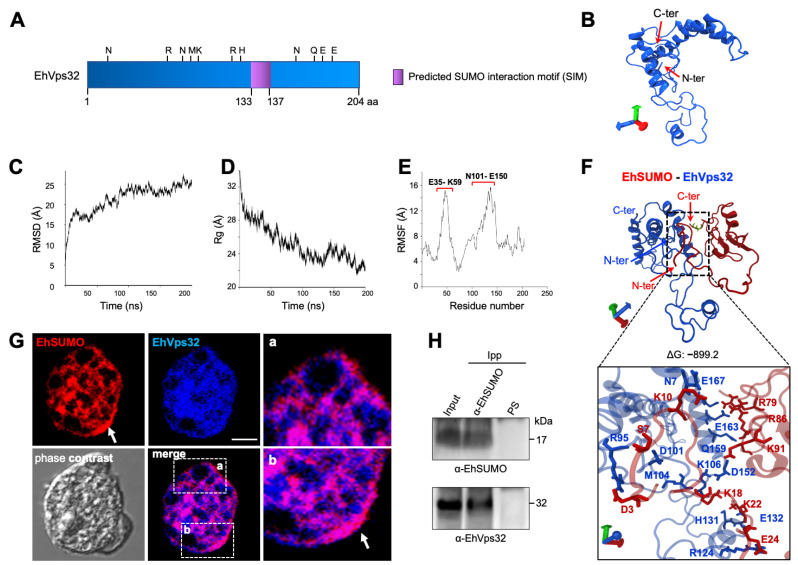
Association of EhSUMO and EhVps32. (**A**) Schematic representation of SIM (purple) site in EhVps32 protein. (**B**) Predicted model of EhVps32 after 200 ns of MDS. The N- and C-terminus regions are indicated. Structural analysis of MDS carried out by (**C**) RMSD, (**D**) radius of gyration, and (**E**) RMSF. Red brackets indicate the most flexible regions. (**F**) Molecular docking of EhSUMO (red) and EhVps32 (blue). Square: magnification of the interacting residues. Sticks in green indicate the glycines in EhSUMO. Axes: x, red; y, green; z, blue. (**G**) Immunofluorescence assays of trophozoites under basal conditions using α-EhSUMO (red) and α-EhVps32 (blue) antibodies. Squares: magnification of proteins colocalization (a, b, and arrow). Arrow: area of colocalization in the internal plasma membrane. Scale bar = 10 µm. (**H**) Trophozoites in basal conditions were lysed and immunoprecipitated using α-EhSUMO antibody or pre-immune serum (PS) and immunoprecipitated proteins were analyzed by western blot using α-EhSUMO or α-EhVps32 antibodies. Numbers at right: molecular weight of immunodetected proteins in western blot assays.

**Figure 9 ijms-22-05709-f009:**
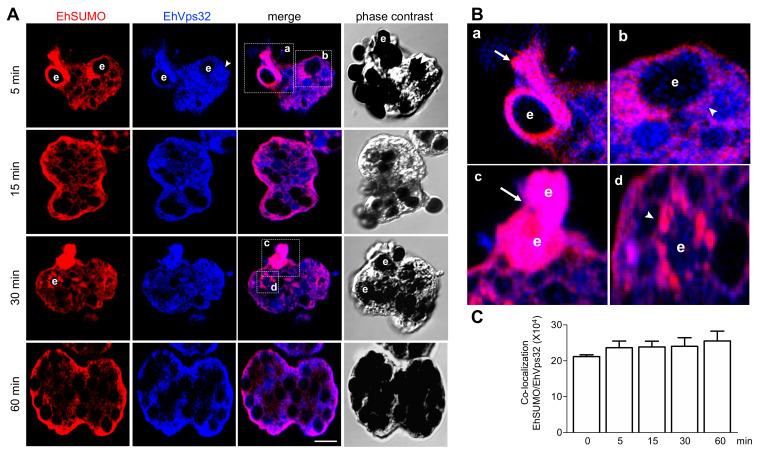
Location of EhSUMO and EhVps32 during phagocytosis. Trophozoites were incubated for 5, 15, 30, and 60 min with erythrocytes (**e**) and processed for confocal microscopy. (**A**) Immunofluorescence assays, using α-EhSUMO (red) and α-EhVps32 (blue) antibodies. Scale bar = 10 µm. (**B**) Magnification of squares presented in (**A**). Arrow: colocalization of proteins in a channel-like structure (**a**) and in erythrocytes (**c**). Arrowhead: colocalization of both proteins surrounding a vesicle with erythrocytes (**b**,**d**). (**C**) Quantification of EhSUMO and EhADH colocalization in the whole cells.

**Figure 10 ijms-22-05709-f010:**
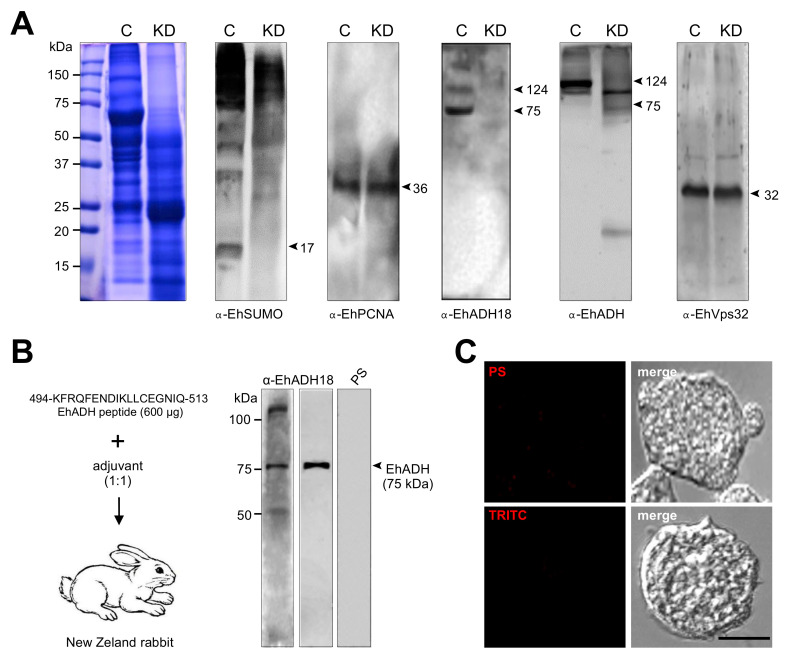
Effect of *EhSUMO* knock-down in the expression of EhADH and EhVps32 proteins. Trophozoites were silenced using the pL4440/*EhSUMO* plasmid as described in materials and methods. (**A**) Coomassie staining or western blot of lysates from control (**C**) or *EhSUMO-KD* trophozoites (KD) in basal condition, using different antibodies: α-EhSUMO, α-EhADH18, α-EhADH, or α-EhVps32. α-EhPCNA: loading control. Numbers at left: molecular weight standards. Number at right: molecular weight in kDa of immunodetected proteins. (**B**) The α-EhADH18 antibody generated in rabbits and using an EhADH peptide of 18 residues, was tested in trophozoites lysates (lane 1) and with the recombinant protein EhADH (lane 2). As a control, the pre-immune serum (PS) was employed in trophozoites lysates (lane 3). (**C**) Confocal images of trophozoites incubated with PS or only with the secondary antibodies (scale bar = 10 µm).

**Figure 11 ijms-22-05709-f011:**
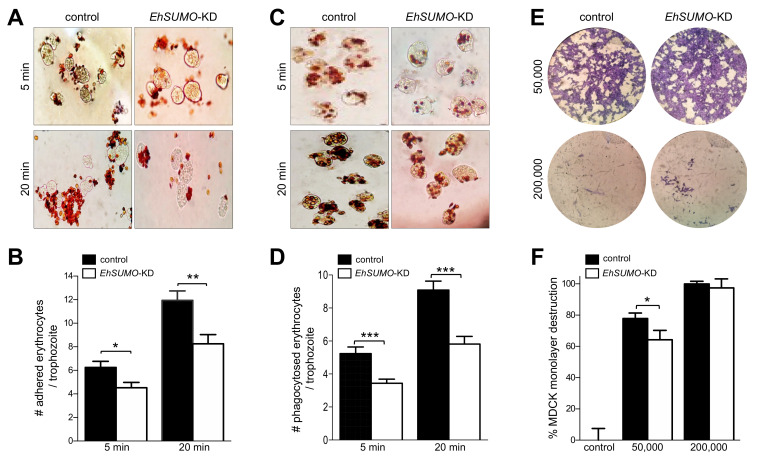
Effect of *EhSUMO* knock-down in the adhesion, phagocytosis, and cytopathic effect. (**A**) Novikoff staining of control and *EhSUMO*-KD trophozoites at 5 and 20 min of adhesion to erythrocytes. (**B**) Number of adhered erythrocytes per trophozoite. (**C**) Novikoff staining of control and EhSUMO-KD trophozoites at 5 and 20 min erythrophagocytosis. (**D**) The rate of erythrophagocytosis was evaluated by counting the number of erythrocytes per trophozoite. Data represent the median and standard error of 100 trophozoites (**E**) The MDCK monolayer damage produced by different amounts (50,000 and 200,000) of control and KD-trophozoites was evidenced by methylene blue staining. (**F**) The destruction of epithelial cells was calculated by measuring the remaining dye concentration in the cell monolayer after contact with trophozoites. Control: MDCK cells non-incubated with trophozoites. Values represent the median and standard error of the three independent experiments. (*) *p* < 0.05, (**) *p* < 0.01, (***) *p* < 0.001.

**Figure 12 ijms-22-05709-f012:**
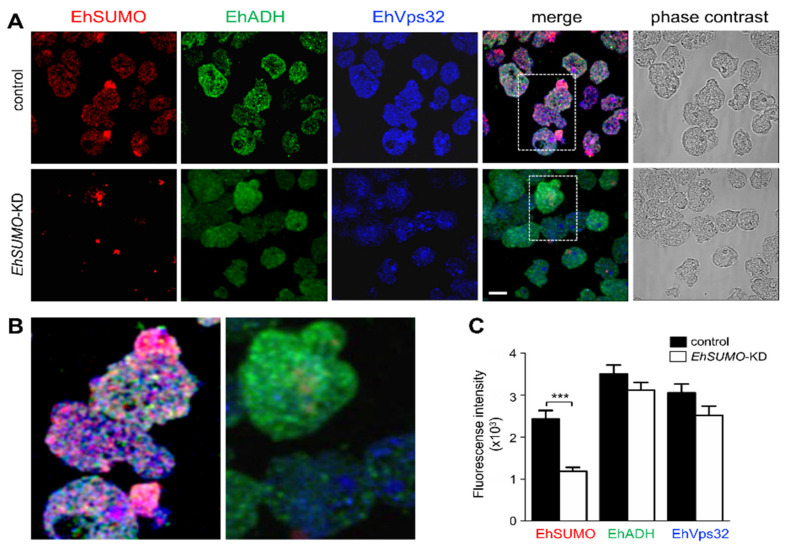
Localization of EhSUMO, EhADH, and EhVps32 in *EhSUMO*-KD trophozoites in basal conditions. Confocal microscopy of control and KD trophozoites in basal conditions. (**A**) Representative image of control and *EhSUMO*-KD trophozoites using α-EhSUMO (red), α-EhADH (green), and α-EhVps32 (blue) antibodies. Scale bar = 10 µm. (**B**) Magnification of squares in (A). (**C**) Fluorescence intensity measured by pixels and corresponding to EhSUMO, EhADH, and EhVps32 in both types of trophozoites. (***) *p* < 0.001.

**Figure 13 ijms-22-05709-f013:**
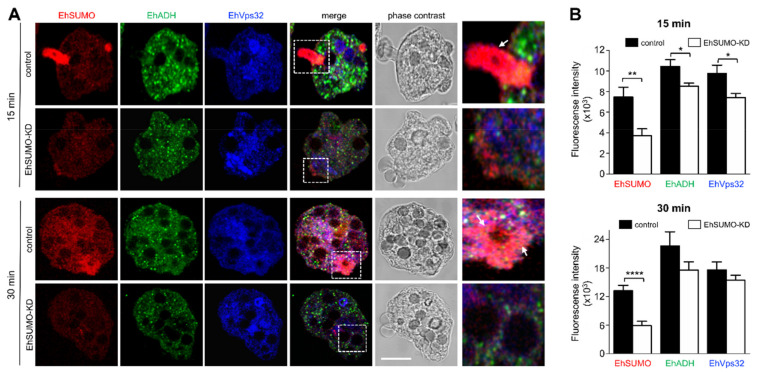
Localization of EhSUMO, EhADH, and EhVps32 in *EhSUMO*-KD trophozoites during phagocytosis. Erythrophagocytosis kinetics of trophozoites silenced using the PL4440/*EhSUMO* plasmid. (**A**) Representative image of control and *EhSUMO*-KD trophozoites at 15 and 30 min of phagocytosis using α-EhSUMO (red), α-EhADH (green), and α-EhVps32 (blue) antibodies and analyzed by confocal microscopy. Scale bar = 10 µm. Right panels: Magnification of squares in merging images. Arrows: phagocytic channel and areas around erythrocytes. (**B**) Fluorescence intensity measured by pixels and corresponding to EhSUMO, EhADH, and EhVps32 proteins in both types of trophozoites at both times of phagocytosis. (*) *p* < 0.05, (**) *p* < 0.01 and (****) *p* < 0.0001.

**Table 1 ijms-22-05709-t001:** *In silico* interactions of EhSUMO with other *E. histolytica* putative proteins related to the SUMO machinery.

Putative Protein	Access Number
Ran GTPase-activating protein	EHI_185290
SP-RING zinc finger domain-containing protein	EHI_069470
SP-RING zinc finger domain-containing protein	EHI_152530
Ubiquitin-conjugating enzyme family protein ubiquitin-conjugating enzyme E2	EHI_147470
Ubiquitin-conjugating enzyme family protein	EHI_178500
Ulp1 protease family, C-terminal catalytic domain containing	EHI_067510
Ubiquitin-activating enzyme ubiquitin-like 1-activating enzyme E1 B	EHI_035540
Ulp1 protease family, c-terminal catalytic domain containing	EHI_097940
Proliferating cell nuclear antigen (PCNA)	EHI_128450

**Table 2 ijms-22-05709-t002:** Comparison of putative proteins involved in SUMOylation in *E. histolytica*, *S. cerevisiae*, *H. sapiens*, and *G. lamblia*.

Protein	*E. histolytica*	*S. cerevisiae*	Identity	*H. sapiens*	Identity	*G. lamblia*	Identity	Protein Type
SUMO	EHI_170060	YDR510W	55%	NP_008868.3	48.35%	GL50803_7760	33%	SUMO protein ubiquitin-activating enzyme, putative
E1	EHI_035540	YDR390C	35%	NP_005490.1	36%	GL50803_6288	25%	SUMO-conjugating enzyme UBC9, putative
E2a	EHI_178500	YDL064W	51%	NP_003336.1	49.67%	GL50803_24068	46%	SUMO-conjugating enzyme UBC9, putative
E2b	EHI_147470	YDL064W	56%	NP_003336.1	53.55%	GL50803_24068	49%	sumo ligase, putative
E3a	EHI_098320	YOR156C	29%	NP_775298.1	27.88%	GL50803_11930	38%	sumo ligase, putative
E3b	EHI_069470	YOR156C	31%	XP_011538282.1	32.22%	GL50803_11930	40%	sumo ligase, putative
* Upl1a	EHI_067510	YPL020C	27%	XP_006719425.1	27.5%	GL50803_16438	25%	sentrin/sumo-specific protease, putative
* Upl1b	EHI_097940	YIL031W	24%	NP_001070671.1	23.6%	GL50803_16438	32%	Ulp1 protease family, C-terminal catalytic domain-containing protein

**Table 3 ijms-22-05709-t003:** Predicted SUMOylation sites of ESCRT-III and EhADH proteins.

ESCRT Complex	Protein	Position	Peptide/Sequence	Score	Site	Protein Size (aa)
**ESCRT-III**	EhVps2	117	RKVNEAT**K**LPAMQKV	24,671	SUMOylation	246
	EhVps20	32–36	VTDLDQKIVDLDRQIRQNI	65,754	SIM	206
	EhVps24	176	EGGIEAV**K**NEVIAES	43,439	SUMOylation	205
	EhVps32	133–137	NNEKSHEIGDLLGEDLQDI	64,005	SIM	204
**ESCRT** **Accessory Proteins**	EhADH	154	QAAGAFQ**K**AADCAQL	25,680	SUMOylation	687
	-	366–370	EYNSKAQ VILND SKKCES	58,036	SIM	-

## Data Availability

The data presented in this study are available in article and Appendix A.

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
