# Peer review of "Protein Sumoylation Is Crucial for Phagocytosis in Entamoeba histolytica Trophozoites"

_ijms, 2021, doi:10.3390/ijms22115709_

Round 1

Reviewer 1 Report

This was a clear, informative and interesting manuscript. There were some concerns about the English style and areas have been highlighted in the text file.

Author Response

Thank you very much to Reviewer 1 for useful comments. We have improved the English style and corrections are highlighted in a copy of the new version

Reviewer 2 Report

Protein SUMOylation is crucial for phagocytosis in Entamoeba histolytica trophozoites

IJMS 2021

Summary:  The authors aim to characterize EhSUMO protein, both in silico and functionally. They present strong evidence that SUMO is present in E. histolytica, and that it is likely involved in erythrophagocytosis, based on localization and knockdown phenotype. However, claims about the importance of interactions with EhADH and Vps32 are overstated. In addition, the value of this work would be increased if the potential for involvement of EhSUMO in other cellular/virulence processes were explored.

Major revisions:

  • A pre-immune serum  control was presented for some experiments, but not for either  IFA or the amoebic lysate western. Considering the importance of the IFA to the conclusions of this manuscript, at least one of these should be shown so that the potential for cross reaction of the serum with other amebic proteins can be judged.
  • Language presenting the results of MDS and other in silico analysis (esp docking modelling) should be improved to make it clearer that it relies solely on modelling and not experimental data. For instance in line 154: "The EhSUMO 3D structure" should be "The putative EhSUMO 3D structure". Predictions of interaction sites between two proteins for which the structures have not been determined experimentally should be similarly caveated.
  • Evidence for changes in EhADH and EhVps32 after EhSUMO knockdown (Figs 11+12) seems weak. Overall changes in fluorescence levels by IFA are minor, and loss of visible co-staining is to be expected considering the reduction in EhSUMO levels, particularly for EhADH which only had very limited co-localization in the wild-type parasites.  If the authors cannot provide additional support for the claims that EhSUMO knockdown alters localization/levels of these proteins they should be softened if not removed.
  • The authors focus on erythrophagocytosis in this study, as that is their main interest. However, the general importance of this paper to the community would be strengthened if the potential for involvement of EhSUMO in other, related, virulence processes (trogocytosis, adhesion, etc) were explored. At the least, a list of all amebic proteins identified as having potential SUMOylation sites should  be included as supplemental data. This would also give context for understanding how common these motifs are.

Minor revisions:

  • Source of ADH18 antibody was unclear - I did not see either a citation or a description of the generation of the Ab in the text. Please provide.
  • Why is there no -RT control for  the RT-PCR? A water control does not distinguish between amplification of cDNA and contamination with genomic DNA.
  • P2 line 89: did you mean 33% identity?

Author Response

Thank you very much to Reviewer 2 for your useful comments.

  1. We have improved the English language and all changes were highlighted in the new version of the manuscript.
  2. We have also improved the description of the material and methods section
  3. We have added the controls required by you in the respective figures.  Pre-immune sera do not recognize any amoebic protein and antibodies are neutralized by the recombinant EhSUMO.   In the case of anti-SUMO antibodies, they recognize many bands because many proteins are SUMOylated.  However,  the recombinant EhSUMO is specifically recognized by the same antibodies as a single band of the corresponding molecular weight in induced bacteria. Additionally, we included for the reviewers and editor the fluorescence pattern obtained with an anti-ubiquitin antibody, because ubiquitin has high similitude with SUMO. Images demonstrate that the fluorescence patterns are completely different (attached file). We have added the western blot controls showing that the antibodies recognize a single band in each case.
  4. Through the text, we have softened the language by adding in each case the word putative or predicted. 
  5.  We have also softened the interpretation of the immunofluorescence figures using anti-EhADH and anti-Eh32 antibodies.
  6.  We have added to the manuscript adherence efficiency and cytopathic effect experiments.  Adherence was slightly inhibited, but we found only a retarded of the monolayer destruction
  7. We have included in supplementary tables a list of published amebic proteins proposed as involved in phagocytosis with the putative SUMOylation sites included as supplemental data.  We agree that this would give context for understanding how common these motifs are.
  8. The source of ADH18 antibody was included in the manuscript in the material and methods section 
  9. We have added extra controls including the -RT control in the RT-PCR experiments.
  10.  We have corrected the line 89.

Round 2

Reviewer 2 Report

Thanks for the revision, especially the inclusion of the controls for IFA. My only question is about the -RT control. In the reply letter you said you had supplied it, but in the new manuscript the control is labeled -DNAse, which is not at all the same thing. Is this a text error or is that the control you did? I am not sure how that is a control for RT-PCR.